# Multitask Approach to Localize Rhizobial Type Three Secretion System Effector Proteins Inside Eukaryotic Cells

**DOI:** 10.3390/plants12112133

**Published:** 2023-05-28

**Authors:** Irene Jiménez-Guerrero, Francisco Javier López-Baena, Carlos Medina

**Affiliations:** Departamento de Microbiología, Universidad de Sevilla, Avenida de Reina Mercedes, 6, 41012 Sevilla, Spain

**Keywords:** type III secretion system, type III-secreted effector, bacterial effector, heterologous expression system

## Abstract

Rhizobia can establish mutually beneficial interactions with legume plants by colonizing their roots to induce the formation of a specialized structure known as a nodule, inside of which the bacteria are able to fix atmospheric nitrogen. It is well established that the compatibility of such interactions is mainly determined by the bacterial recognition of flavonoids secreted by the plants, which in response to these flavonoids trigger the synthesis of the bacterial Nod factors that drive the nodulation process. Additionally, other bacterial signals are involved in the recognition and the efficiency of this interaction, such as extracellular polysaccharides or some secreted proteins. Some rhizobial strains inject proteins through the type III secretion system to the cytosol of legume root cells during the nodulation process. Such proteins, called type III-secreted effectors (T3E), exert their function in the host cell and are involved, among other tasks, in the attenuation of host defense responses to facilitate the infection, contributing to the specificity of the process. One of the main challenges of studying rhizobial T3E is the inherent difficulty in localizing them in vivo in the different subcellular compartments within their host cells, since in addition to their low concentration under physiological conditions, it is not always known when or where they are being produced and secreted. In this paper, we use a well-known rhizobial T3E, named NopL, to illustrate by a multitask approach where it localizes in heterologous hosts models, such as tobacco plant leaf cells, and also for the first time in transfected and/or *Salmonella*-infected animal cells. The consistency of our results serves as an example to study the location inside eukaryotic cells of effectors in distinct hosts with different handling techniques that can be used in almost every research laboratory.

## 1. Introduction

Rhizobia are a group of soil bacteria that, in natural conditions, live in a saprophyte way. However, they can also establish symbiosis with legume roots and proliferate inside them when these plants are under nitrogen starvation. In such conditions, rhizobia induce the formation of nodules, plant organs that will host rhizobia in the legume root cortex, in response to specific bacterial signals. Once inside nodules, rhizobia differentiate into bacteroids—an endosymbiotic form of these bacteria able to reduce atmospheric dinitrogen to ammonia that will be supplied to the plant—which in turn provide bacteria with different carbon sources and a safe environment [1].

The interaction of many bacteria with eukaryotic hosts is mediated by bacterial outer proteins secreted through different secretion systems. Such proteins are released to the extracellular milieu where they interact with the surface of the eukaryotic cells or can also be directly delivered into the cytosol of the host cell through type III, IV, or VI secretion systems. Rhizobia, like many other plant-interacting bacteria, use distinct protein secretion systems to translocate proteins to the legume root cells, which contributes to the invasion of plant tissues. Moreover, the role that the lipochitooligosaccharides (LCO) play in the development of legume nodules has been broadly analyzed [2,3]. In addition to such molecules, the involvement of rhizobial surface polysaccharides in the colonization of the roots is also important in symbiosis since they can suppress the defense responses of the plant, allowing the penetration of the bacteria [4,5]. However, only recently has the study of rhizobial secreted proteins been gaining importance due to the large number of functions that they modulate in legumes during the root colonization process, on one hand contributing to the suppression of the plant immune response and, on the other, participating in the specificity of the relationship between rhizobia and their natural hosts [6].

Among the protein secretion systems present in bacteria, the type III secretion system (T3SS) has been the most studied and is well understood in phytopathogenic bacteria [7,8,9]. Proteins secreted by this secretion system, named type III-secreted effectors (T3E), have evolved to counteract the plant immune response in such a fashion that they can manipulate different plant cell targets, hampering the defensive responses. Thus, T3E interfere with many cellular processes such as secretory pathways, cytoskeleton polymerization, transcription or translation processes, and cell signaling [10,11]. In the case of pathogens, T3E have a clear role in causing damage to the host cell. During the evolutionary never-ending race among plants and phytopathogens, while bacteria have raised new weapons (T3E) to overcome plant defenses and invade their tissues, plants have reinforced their resistance by developing new elements (R proteins) that recognize the T3E to neutralize them [12]. Curiously, the symbiotic rhizobial T3SS has two possible effects on the establishment of the legume symbiosis, which is strictly related to the specificity of the plant–bacterium relationship. In some cases, symbiotic T3E are recognized by plant protein receptors that block nodulation, determining host specificity, which resembles the plant resistance described for phytopathogens. However, in other cases, the T3E are indispensable in inducing the nodulation process in some specific plant cultivars, arguing some clues in the co-evolution of this symbiotic relationship. The double role of the rhizobial T3SS in symbiosis has been recently reviewed [6,13].

While phytopathogens have evolved a huge number and diversity of T3E to counteract plant R proteins, rhizobial strains have a much lower number. This is the case of *Sinorhizobium fredii* strain HH103 (hereafter HH103), with only eight T3E whose function in symbiosis has been determined for some hosts [14,15]. Although this double role in symbiosis is relatively easy to determine, just checking the ability of wild-type strains and mutants in the T3SS or in certain T3E to nodulate different plant hosts, the molecular role of the effector is not as trivial to define. The first step in determining whether a protein is indeed a T3E is the confirmation of its translocation through the T3SS. The enzymatic methods based on CyaA cyclase activity or glycogen synthase kinase (GSK) tags [16,17] are useful for demonstrating this translocation, but they present some limitations, such as a spatio-temporal lack of information, since the whole population of eukaryotic cells is analyzed and the knowledge of where or when the effectors act is missed. For these reasons, these methodologies must be complemented with microscopic approaches that demonstrate translocation and localization of T3E within single host cells [18].

To contribute to previous microscope procedures in the analysis of bacterial T3E, we combined different existing methodologies in the visualization and localization of the rhizobial effector NopL. This effector has been deeply studied in *S. fredii* NGR234 (hereafter NGR234) and shows a high level of identity with NopL from HH103 (96% identity). The NGR234 NopL is phosphorylated by *Nicotiana tabacum* and *Lotus japonicus* kinases. Furthermore, by using tobacco protein extracts, it was shown that the kinase involved could be a serine/threonine MAPK-related kinase [19]. Subsequent studies described that NGR234 NopL possesses 4 phosphorylable serine residues in its protein sequence, whose serine/threonine pattern is typical for MAPK substrates [20]. Likewise, other studies point out that the function of the NopL effector could be the modulation of plant defense signaling pathways, mediated by MAPK, that culminate in the activation of *PR* genes [21]. More recently, transient expression of NopL in tobacco leaves has been reported to reduce the effects of the salicylic acid-induced protein kinase (SIPK), whose expression is induced by high levels of salicylic acid (SA) in the presence of a pathogen [20].

One of the attributed functions for SIPK is the induction of nodule senescence through oxidative stress reactions that can be suppressed by NopL, therefore expanding the lifespan of infected host cells, and hence extending the nitrogen fixation in nodules [20,22]. Fluorescence microscopy techniques allowed the reasoning that such phosphorylation reactions occur in the nucleus, where NopL is targeted and complexed to SIPK, as demonstrated in epidermal onion cells or in the leaves of *N. tabacum* plants. These approaches, although carried out in heterologous hosts, are needed to contribute to the interpretation of the role of NopL in a particular subcellular location inside host cells due to the inherent difficulties that legume roots have for the use of fluorescence microscopy.

Since it has been well described that NopL interferes with the MAPK signaling pathway, a conserved pathway in higher eukaryotes [23], and the protein is targeted to the plant cell nucleus [22], here we propose the use of alternative heterologous hosts to confirm that other models and different methodologies can be used to analyze the subcellular location of the HH103 effector NopL, namely (i) transient expression of NopL in *N. benthamiana* leaves mediated by agroinfiltration with *Agrobacterium tumefaciens*, (ii) *Salmonella*-induced overexpression of NopL and delivery into the cytosol of HeLa-infected cell cultures, and (iii) transient expression of HeLa cells transfected with NopL coding DNA cloned into eukaryotic vectors. The results obtained are consistent in every case and contribute to the investigation of the subcellular location of rhizobial T3E in heterologous hosts.

## 2. Results

### 2.1. The S. fredii HH103 NopL Is Phosphorylated by Soybean Kinases

Comparative analysis of amino acidic sequences revealed that the HH103 NopL (338 aa, YP_006575344) shares 96% of identity with the corresponding NopL protein from NGR234 (338 aa, NP_444148) [15]. This effector protein is a substrate of plant kinases and phosphorylable residues are conserved between NGR234 and HH103 NopL proteins [19,20]. To determine whether kinases from the HH103 natural plant host, soybean, can also phosphorylate the HH103 NopL, in vitro phosphorylation assays were carried out. For this purpose, plasmid pMUS1135 was overexpressed in *E. coli* BL21 (DE3) to purify the HH103 NopL protein (Appendix A). In parallel, extracts from soybean roots were obtained from uninoculated plants or from plants inoculated with HH103 and harvested 4 days post-inoculation to determine whether the involved kinase(s) is/are constitutively expressed or are inducible by the presence of a compatible rhizobial strain.

As expected, radioactive signals corresponding to phosphorylated NopL-GST were detected in the gels. However, no significant differences in the intensity of the signals were observed in the presence of inoculated or uninoculated soybean root extracts (Figure 1a). In all cases, the presence of Ca^2+^ in the phosphorylation reaction was essential for the successful phosphorylation of NopL [24,25,26]. No radioactive signal was detected at the molecular weight corresponding to GST, suggesting that GST was not phosphorylated by soybean extracts (data not shown). As controls, protein extracts from uninoculated or wild-type inoculated roots were independently assayed to demonstrate that the signal detected is not due to the phosphorylation of the proteins present in these extracts. Therefore, no phosphorylation signal was observed in these treatments (data not shown).

To elucidate whether the NopL-phosphorylating kinases from both NGR234 and HH103 possess the same nature, in vitro phosphorylation assays using different kinase inhibitors were performed. These included genistein (a tyrosine kinase inhibitor), K252a (a serine/threonine kinase inhibitor), KN62 (a CaM kinase II inhibitor), PD98059 (a MAP kinase kinase inhibitor), EGTA (a Ca^2+^ chelator), and N-(6-aminohexyl)-5-chloro-1-naphthalene sulfonamide (W5 and W7, calmodulin antagonists). Inhibition effect was considered when the phosphorylated NopL signal was less than 50% with respect to the control. As shown in Figure 1b, NopL phosphorylation was reduced by K252a and PD98059, which is consistent with the data previously described for the NGR234 NopL [19]. The inhibitor genistein did not substantially alter the phosphorylation of NopL. In addition, the use of inhibitors as calcium-related compounds resulted in a significant reduction in the NopL phosphorylation signal, indicating that the soybean pathway that culminates in the phosphorylation of NopL is calcium- and calmodulin-dependent and that a calmodulin (CaM) kinase II can be directly or indirectly involved, as previously described [24,27]. Thus, once it was demonstrated that the HH103 NopL shares the same phosphorylation characteristics as the NGR234 NopL, we analyzed the HH103 NopL subcellular localization in different heterologous hosts.

### 2.2. Subcellular Localization of NopL in Tobacco Leaves

Transient expression in heterologous host plants has been broadly used to establish the subcellular localization of bacterial effectors. To transfer the effector encoding genes to plant cells from distinct species and/or tissues, cells can be bombarded with particles containing eukaryotic replicative vectors bearing fusions of effectors with fluorescent genes [22]. Independently, agroinfiltration mediated by *A. tumefaciens* in tobacco leaves has been demonstrated to be an efficient method to achieve transient expression of heterologous proteins in plant hosts [28]. Among the tobacco plants used, *N. benthamiana* is highlighted due to its easy handling in co-infiltration experiments with *A. tumefaciens* and the large number of infiltratable leaves prone to produce high levels of heterologous proteins that prevent excessive necrosis after Agrobacterium infiltration [29].

To determine the localization of NopL in tobacco, *N. benthamiana* leaves were infiltrated with *A. tumefaciens* carrying vector pMUS1250, which added a YFP fused to the C-terminus of NopL, with pEarleyGate 104 as a positive control, which constitutively expresses YFP, and pEarleyGate 100 as a negative control. Infiltrated leaves were visualized after 48 h. As previously described in onion epidermal cells [21], NopL-YFP fusions were mainly detected as nuclear dots in the plant nuclei, while the YFP in the control treatment was distributed throughout the cytoplasm and the nucleus (Figure 2). Several leaves from independent plants were infiltrated with each condition and numerous cells with yellow spots in nuclei were observed.

### 2.3. NopL Overexpression by Salmonella and Detection on Infected HeLa Cell Cultures

To check the possibility of analyzing the subcellular location of NopL in a different heterologous system of simpler control, we infected HeLa cells with Salmonella strains harboring a protein overexpression system that drives the expression of NopL. This system allows the tight control of heterologous protein expression upon the addition of the inducer molecule salicylate, and its performance has been validated in Salmonella infecting different hosts [30,31]. Briefly, the system consists of a regulatory module integrated into the chromosome that is composed of regulatory elements from distinct Pseudomonas strains. This part of the system links two divergent promoters that activate the expression of two proteins responsive to salicylate, NahR and XylS2 [Pnah/nahR:Psal/xylS2]. On the other hand, another part of the system loaded in a plasmid includes the expression module that drives the transcription of a gene of interest from the Pm promoter which is activated by the XylS2-salicylate activated protein. This way, the amplification of the transcriptional activation allows the overproduction of a protein that otherwise should be hard to detect [30].

Since the T3SS has a common ancestor, its needle-like basic structure, called an injectisome, is relatively conserved among most eukaryotic-interacting bacteria holding this system. Seven families of injectisomes have been identified so far through phylogenetic analysis, showing a completely different evolutionary tree from other trees based on 16S rRNA [32]. Even though the T3SS of Salmonella and Sinorhizobium belong to different families (Ssa-Esc and HrcN-RhcII, respectively [32,33]), and following previous results of effector secretion through heterologous type III machinery [34], we tried to use the salicylate-induced cascade expression system to overexpress and putatively secrete the HH103 NopL to HeLa cell cytoplasm through the Salmonella T3SS (Figure 3). To visualize the location of the protein, we fused an HA epitope to the C-terminus of the HH103 NopL to immunodetect it by fluorescence microscopy while inducing its expression from the Pm promoter, as described in Section 4. As observed in Figure 3, the HH103 NopL-HA, once produced in Salmonella for 4 h, could not be translocated across the Salmonella T3SS and the fluorescence remained confined to the bacteria, while in the positive control of secretion, the amino-terminal peptide of the Salmonella effector SspH2, fused in frame to HA, was spread in the host cytosol, demonstrating an effective translocation, as previously reported [30].

However, we used another approach to release NopL produced by Salmonella in the HeLa cells cytosol, consisting of a bacterial-induced autolysis system based on the lambda phage lysis mechanism that, in response to anhydrotetracycline (AHT), induces the lysis of Salmonella [35]. In short, this system combines the AHT inducible tet promoter with the lambda lysis gene cluster SRRz, whose expression derives from the lysis of the bacterial membrane and the degradation of the peptidoglycan [36,37]. This lysis system is contained in a plasmid compatible with the previous one that drives NopL-HA induction, and both were introduced in a Salmonella sifA mutant strain that lacks the Salmonella surrounding vacuole (SCV) and therefore allows the bacteria to live free in the eukaryotic cytosol as an intracellular protein factory [35,38]. Once NopL-HA was overproduced upon salicylate induction as previously described, the bacterial content was released in the cytosol of HeLa cells after the addition of AHT, and the cells were visualized about 24 h after the lysis induction. In this case, we immunodetected the accumulation of NopL mainly inside the nuclei concentrated in nuclear dots, while in the absence of the autolysis system coupled to the overproduction system, the NopL-HA remained inside the bacteria (Figure 4).

### 2.4. NopL Transient Expression and Detection on Transfected HeLa Cells

To exclude the possibility that the results obtained in cell cultures are artefacts derived from the Salmonella infection, a transient expression experiment was carried out in HeLa cells. The transfection of animal cell cultures with the effector coding DNA cloned into eukaryotic expression vectors has been used to analyze the role of a single pathogen effector on host cells to avoid the masking of its effect when other effectors are present in a conventional infection experiment [39]. According to this, we cloned the NopL encoding gene to the C-terminus of the Venus green fluorescent reporter [Venus-NopL] in a derivative of the plasmid vector pcDNA 5/FRT/TO that contains a hybrid promoter composed of a fragment from the Citomegalovirus promoter fused to the tetracycline operator, which makes it responsive to AHT. This vector was introduced into HeLa cells by lipofectamine-mediated transfection, and the expression of the fusion protein was induced by the addition of AHT. After 24 h, the cell culture was visualized and Venus-NopL was localized mainly in the HeLa cell nuclei concentrated in nuclear dots, as detected in the above sections (Figure 5). The transient expression was tracked for 48–72 h, showing that during this time, the NopL protein, although present in the HeLa cytoplasm, was mostly accumulated in the nuclei (bottom panels). In contrast, in the empty vector used as a negative control, the green fluorescence was uniformly spread in the cytoplasm and nuclei during the experiment (upper panels), confirming that NopL is localized in the nuclei of HeLa cells even when self-expressing it.

To check whether the accumulation of NopL in the nucleus of HeLa cells exerted any effect on the cell cycle progression or the cell viability, we monitored it by flow cytometry analysis for 72 h. Unfortunately, no visible differences in population peaks’ distribution were appreciated when the cells were transfected with Venus-NopL in comparison with those transfected with the empty vector, with just a slight increase in the subG1 population after 72 h in the NopL-expressing cells (*p* = 0.057) that represent the dead cell population.

## 3. Discussion

Bacterial effector proteins have evolved to reach eukaryotic cell cytosol through different secretion systems such as type III, IV, or VI and accomplish their biochemical function in such environments. In the first phase, the effectors are injected through needle-like structures into the eukaryotic cytosol to subsequently be targeted to a precise subcellular compartment where they can increase their effective concentration to exert their function. There are multiple examples of the destinies that effectors take inside the eukaryotic cells as cytosol, membrane, mitochondria, chloroplasts, proteasome, or nucleus, and of course, there are many well-documented cellular processes that they modulate, including membrane trafficking, cytoskeletal dynamics, energetic fitness, signal transduction, and many others. Many brilliant reviews discuss the role of bacterial effectors in their precise cellular target compartments [40,41].

After the post-genomic era, the identification of bacterial effectors in genomes is becoming feasible by means of machine learning approaches that can predict new effectors based on the similarity of their DNA sequence to known effectors, or even calculating the probability of a given protein to be an effector [42,43]. Among these bioinformatic tools, some of them are even able to predict the precise subcellular compartment where the effector would be targeted, based on the presence of eukaryotic-like protein domain signals that determine their final objective.

However, despite this large genome-scale analysis that provides valuable insights on DNA sequences, the validation of the predicted effectors is mandatory. Once the T3E host cell translocation is verified, as an initial analysis of the effector activity, the determination of its subcellular target(s) contributes to the description of their function. In fact, there exist several methods to monitor the bacterial effectors’ trafficking inside the eukaryotic cells, most of them based on the fusion of the hypothetical effector to a reporter tag of a different nature [18,44,45]. These systems aim to counteract the lack of specific antibodies for every protein or secretory domain to be studied and have the great advantage of amplifying the signal of poorly expressed effectors by using secondary antibodies. Although these novel methodologies (FAST, NanoLuc, LOV...) support researchers with additional tools, traditional fluorescence-based methods to track bacterial effectors in vivo are still extensively used due to their consistency, despite several intrinsic limitations such as the possibility of effector misfolding or translocation precluding during the secretion through its corresponding system.

In the specific case of rhizobacterial effectors, the tracking of its translocation to the root cells of their natural hosts is not always feasible due to the root autofluorescence and the inherent difficulty of handling each cognate partner of such soil bacteria. To solve this limitation, many plant models such as Arabidopsis thaliana or Lotus japonicus have been used due to the good results obtained when they are infected with bacteria and their roots visualized under confocal or epi-fluorescence studies [46,47,48]. A simpler approach to this natural condition relies on the agroinfiltration of *N. benthamiana* leaves mediated by *A. tumefaciens* to reach the transient expression of a bacterial effector in plant cells that can be easily visualized under confocal fluorescence, avoiding the fixation of the sample [49]. This robust technique has been broadly used to detect the presence of phytopathogen effectors in subcellular compartments [50].

In this study, we are focused on the well-known rhizobial T3E protein NopL. Here, we determined that NGR234 and HH103 NopL are highly homologous, since their aminoacidic sequences present high identity and, due to both proteins, are phosphorylated by a serine/threonine plant MAPK-related kinase (Figure 1) [19]. Following the approach described above, we fused NopL to a YFP and detected the yellow fluorescence mainly in the plant cell nucleus, and, more specifically, concentrated in nuclear dots where they probably interact with tobacco nuclear MAP kinases, as previously described by [22] using onion epidermal cells (Figure 2). Additionally, we checked alternative models to analyze the subcellular destiny of bacterial effectors using the rhizobial NopL as a model. Thus, we could determine whether this effector seems to be distributed in a similar way by other heterologous systems not related to the natural conditions of Sinorhizobium. It has been previously reported that the rhizobial effector NopM is an E3 ubiquitin ligase belonging to the IpaH effector family found in Shigella, Salmonella, and Yersinia, all of them sharing the same function: the transference of ubiquitin to proteins to be tagged for degradation in the proteasome. This ubiquitin-mediated proteasome-dependent protein degradation is conserved in eukaryotic cells, and yeasts have been used as eukaryotic models to analyze the role of bacterial effectors that modulate its function regardless of whether the effector is from plant- or animal-interacting bacteria [51].

Because NopL seems to be phosphorylated by a MAPK-related kinase, and taking into consideration the conservation of MAP kinase signaling pathway cascades in higher eukaryotes [23], we selected HeLa cells to study NopL location expressed from different sources. The striking parallels between the in-host lifestyles of rhizobial strains and other mammalian pathogens, in addition to the fact that human HeLa cells are a widely used experimental infection model even for plant-interacting bacteria [52,53], led us to use this model for which we had solid experience and many molecular tools available. In fact, even though HeLa cells are not the natural hosts of rhizobial strains, they have been previously used as heterologous hosts to analyze the cytoskeletal reorganization that suffers when they are infected with such bacteria [52].

Among other difficulties in the study of the role of bacterial effectors inside mammalian cells, the low concentration to which the effectors exert their function and their local effects limit the depth of the knowledge that can be acquired from them. To cope with this limitation, different protein overexpression systems have been used to magnify the concentration of effectors inside the host cells. In this regard, we have previously contributed to the study of the Salmonella effector SpvB by means of the salicylate cascade expression system [54]. This protein is produced by Salmonella when it infects eukaryotic cells at physiological levels to inhibit the re-polymerization of actin monomers, and as a result, Salmonella can control the vesicle trafficking to survive and proliferate intracellularly. The low concentration at which this protein operates inside the cell has restricted its further characterization, and the synthetic alteration of its transcription was sufficient to gain insight into its functioning system [54]. In the same direction, we have overexpressed NopL from Pm promoter in different Salmonella strains to putatively detect a high concentration of this protein in a determined subcellular compartment of HeLa cells. In a first attempt and following previous evidence that showed the secretion of a chimeric protein of Chlamydia spp. by the Shigella flexneri T3SS machinery [34], we tried the heterologous secretion of NopL through S. typhimurium T3SS. Unfortunately, we were not able to detect the NopL-HA epitope in HeLa cytosol by immunofluorescence assays through this approach (Figure 3). Therefore, we used a different procedure that we previously used to release heterologous proteins overproduced by Salmonella into HeLa cytosol. This system combines the overexpression of a given protein via salicylate induction, with a controlled system to induce the autolysis of Salmonella, which is activated only when the amount of the protein is raised upon salicylate induction. In this case, we could detect NopL inside the HeLa cell nuclei and precisely concentrated in nuclear dots, with a minimal presence of the protein inside some bacteria (Figure 4), in which the autolysis was probably incomplete, as we previously documented [35]. Thus, in agreement with the previous approach in *N. benthamiana* leaves, we show that using this alternative method, the nuclear detection of NopL is equivalent.

Nevertheless, we have complemented this study with the effector transient expression in transfected HeLa cells, which is a more similar approach to that expressing the T3E on *N. benthamiana* leaves in terms of gene expression. In this case, we have chosen a eukaryotic vector that drives the expression of a heterologous gene, NopL fused to Venus green fluorescent protein (GFP) encoding gene, from the AHT inducible Cytomegalovirus promoter. Following an effective transfection of the cell population with the plasmid vector containing the fusion, which was observed by the green fluorescence of a representative population of the cell culture (not shown), the cultures were traced for 72 h. After 24 h of expression, the green fluorescence, although partially spread in the cytosol, was accumulated in the nuclei of transfected cells in contrast to cells transfected with the empty vector, in which the fluorescence was uniformly distributed throughout the cell (Figure 5). As in the previous approaches, the fluorescence inside the nucleus was also focused on nuclear dots, probably due to the interaction with MAP kinase homologs with SIPK of tobacco, but whose nature is beyond the scope of this study. Hence, we report another possible methodology to detect the subcellular localization of a bacterial effector and contribute to its analysis.

In summary, we have combined different techniques to successfully detect the rhizobial T3E NopL protein in distinct heterologous systems. NopL has been identified as a substrate for nuclear MAP kinases. In addition, this T3E can be hyperphosphorylated by such proteins competing with their natural substrates, consequently modulating the MAPK signaling cascade [19,20,22]. The diverse modes of NopL expression used in this study yield consistent results and support the role of this T3E in the nuclei. The mechanism of NopL nuclear importation remains elusive, even with the use of different expression tools. Thus, although the consistency of these methods points out that this mechanism could be equivalent in animal or plant cells, further studies should be carried out to confirm this observation.

## 4. Materials and Methods

### 4.1. Bacterial Strains and Growth Conditions

Bacterial strains and plasmids used in this work are listed in Appendix A. *S. fredii* HH103 Rif^R^ was grown at 28 °C on tryptone yeast (TY) medium [55]. *Salmonella* and *Escherichia coli* strains were grown at 37 °C and *Agrobacterium tumefaciens* at 28 °C on Luria Bertani (LB) medium and supplemented with antibiotics when necessary [56].

### 4.2. Molecular Biology General Procedures

All DNA manipulations were performed following standard protocols [56], and each cloning is detailed in the section where they are used to avoid duplicities.

### 4.3. In Vitro Phosphorylation Assay

The HH103 effector NopL was phosphorylated in vitro by soybean root kinases following the methodology described by [6], with some modifications. Briefly, 0.2 g of the upper third of the soybean roots grown in hydroponic conditions [7] was ground in a mortar with washed sand and 2 mL of 20 mM Tris-HCl pH 7.5, 100 mM NaCl, 5 mM EDTA, and 1 mM EGTA extraction buffer. Then, the mixture was centrifuged at 12,000× *g* for 5 min at 4 °C, and proteins from the supernatant were precipitated by the addition of (NH_4_)2SO_4_ (0.39 g ml^−1^). After centrifugation at 10,500× g for 5 min at 4 °C, the pellet was completely dried and resuspended in the buffer indicated above supplemented with 2 µL of a cocktail of protease inhibitors (Sigma Aldrich, St. Louis, MO, USA). Proteins were filtered using a Sephadex G-25 column and quantified using the method of Bradford [4] employing BSA as standard. A total of 20–40 μg of these protein extracts was used for the in vitro phosphorylation assays, which were performed with 5–10 μg of the fusion protein in the presence of 1 µCi [γ-32P]-ATP, 1 mM MgCl_2_, and 1 mM CaCl2 for 90 min at 30 °C, followed by SDS-PAGE separation and autoradiography (Fujifilm LAS-5100, Fujifilm, Tokyo, Japan).

The kinase inhibitors PD98059, KN62, and K252a were dissolved in DMSO. Genistein was dissolved in ethanol. The control reaction contained 20% (*v*/*v*) DMSO. The concentrations of the inhibitors used in the assays were 2 mM for EGTA W5 and W7, and 500 µM for PD98059, KN62, K252a, and genistein. Proteins were isolated in their native form to preserve their phosphorylation substrate capacities, and the same volume for each reaction was loaded in each well.

### 4.4. Transient Gene Expression in Nicotiana benthamiana Leaves and Confocal Imaging

The HH103 *nopL* gene without its end codon was amplified using primers nopL_attB1 and nopL_attB2ns (Appendix A). The amplified DNA fragment was cloned into the Gateway donor vector pDONR207 (Invitrogen, Waltham, MA, USA) and then subcloned into the Gateway destination vector pEarleyGate 101 [57]. Vector pEarleyGate 101 adds an in-frame YFP-tag followed by an HA-tag to the C-terminus of the protein. Plasmid pMUS1250 was then transformed into *A. tumefaciens* GV3101, as described by [58].

Transient expression and co-expression assays were performed following the protocol described in [58]. Functional fluorophores were visualized in the infiltrated leaves using a Leica SPE Confocal Microscope (Leica Microsystems, Wetzlar, Germany) 48 h post-infiltration. DAPI (4′,6-diamidino-2-phenylindole) was used at 1 mg mL^−1^ to stain the nucleus of the plant cells [59]. *A. tumefaciens* GV3101 derivative strains carrying pEarleyGate 104 and pEarleyGate 100 were used as positive and negative controls, respectively.

### 4.5. In Vitro Bacterial Infection of HeLa Cells

The NopL gene was PCR-amplified without its end codon from genomic DNA of HH103 with primers NopL Fw/*Nde*I and NopL Rev/*Sal*I (Appendix A) to be cloned under the control of Pm promoter in plasmid pMPO1003 containing the HA epitope in the c-terminus generating plasmid pMPO1617. HeLa cells were grown at 37 °C and 5% CO_2_ on tissue culture plates (Nunc, Roskilde, Denmark) in Dulbecco’s modified Eagle’s medium (DMEM; Sigma) supplemented with 2 mM L-glutamine and 10% fetal calf serum (FCS) containing a penicillin–streptomycin mixture. Cell infections were performed as described elsewhere with minimal modifications [30]. Cells were cultured in 24-well plates at a density of 10^5^ cells per well 20 h before infection. An overnight *Salmonella* culture was diluted 1:33 on fresh LB supplemented with plasmid markers when necessary, and incubated at 37 °C for 3.5 h. For infection, bacteria were added at a multiplicity of infection (m.o.i.) of 100:1, and the infection was allowed to proceed for 15 min at 37 °C and 5% CO_2_. Wells were washed twice with phosphate-buffered saline (PBS) and incubated for 1 h with DMEM containing 100 μg mL^−1^ gentamicin (PAA laboratories GmbH, Austria) to kill extracellular bacteria. Following this, the antibiotic concentration was reduced to 16 μg mL^−1^ to keep the intracellular bacterial population alive during protein expression assays.

### 4.6. Heterologous Protein and/or Lysis Induction in Cellular Infected Cultures

Heterologous protein overexpression was induced from the cascade expression system for 5 to 10 h with sodium salicylate 2 mM (Sigma-Aldrich, Germany). When necessary, the bacterial lysis module was induced with 0.2 μg mL^−1^ of AHT (Clontech Laboratories), and lysis expression was allowed for 13–18 h until analysis.

### 4.7. Fluorescence Immunostaining of Infected HeLa Cells

For fluorescence staining, infected cells were grown on glass coverslips (12 mm, Thermo Scientific, Waltham, MA, USA) and NopL::HA production and/or bacterial lysis was induced as mentioned above. Cell samples were then rinsed twice with PBS, fixed in 3.7% paraformaldehyde for 10 min at room temperature, permeabilized in 0.1% Triton X-100 and/or lysozyme for 10 min, and incubated for 45 min with blocking buffer (3% FCS in PBS) to inactivate HeLa membrane receptors. Thereafter, cells were washed twice with PBS and incubated overnight with 1:250 anti-HA primary antibody (COVANCE) in blocking buffer at 4 °C. Cells were then washed and incubated for 90 min with 1:300 dilution in PBS of anti-mouse IgG conjugated to Alexa555 or Alexa488 secondary antibody (Molecular Probes/Invitrogen) at 37 °C. Subsequently, cells were washed twice with PBS and incubated for one hour with PBS containing Hoechst 33,258 (1 μg mL^−1^) for nuclei staining and/or 1:100 rhodamine phalloidin (R415; Invitrogen Molecular Probes, Eugene, OR, USA) for acting staining when necessary at room temperature in the dark. After washing with PBS, the coverslips were mounted on slides and visualized with a confocal microscope, Leica SPE (630X) (Leica Microsystems GmbH, Wetzlar, Germany). Translocation was considered positive when the fluorochrome-conjugated secondary antibody (red or green) was clearly visible in the cytoplasm or nuclei.

### 4.8. Transfection of Cell Cultures and Cell Cycle Analysis

The NopL gene was PCR-amplified from genomic DNA of HH103 with primers NopL Fw/BamHI and NopL Rev/NotI (Appendix A) for its cloning in plasmid pcDNA5/FRT/TO-Venus-Flag (865) (addgene) in frame with Venus protein to generate plasmid pMPO1643. PCR was performed with high-fidelity polymerase (Roche) as directed by the manufacturer.

To analyze HeLa cells’ transient expression, cells were transfected using the Lipofectamine (Invitrogen) method with 1–2 μg of the DNA constructs pMPO1643 or the empty vector pcDNA5/FRT/TO-Venus-Flag (865). After 4 h of cell incubation with Opti-Mem medium and Lipofectamine, the medium was changed to fresh DMEM containing 20 μg mL^−1^ of AHT to induce proteins from pCMV promoter. The cultures were visualized up to 72 h and pictures were taken when green dots were observed in the nuclei with an inverted fluorescence microscope, Leica DMI4000B (320X) (Leica Systems, St. Gallen, Switzerland). Cell cycle distribution was determined by flow cytometry of propidium iodide (PI)-stained nuclei at 24, 48, and 72 h after induction, as described in [54]. In total, 10,000 events for each sample were analyzed by flow cytometry using CellQuest software to determine the relative DNA content based on the presence of red fluorescence (PI) to evaluate the percentages of cells in sub-G1 (apoptotic cells), G0/G1, S, and G2/M phases. Statistical analysis was performed using Student’s *t* test. *p* < 0.05 was regarded as statistically significant.

## Figures and Tables

**Figure 1 plants-12-02133-f001:**
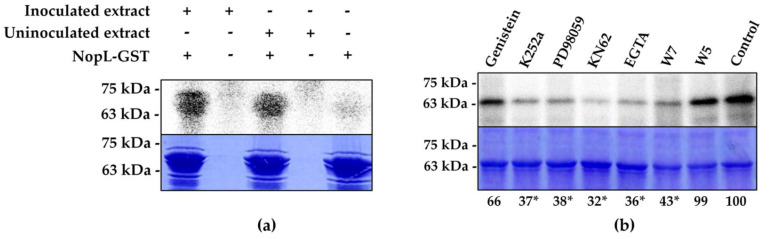
**Figure 1. Soybean root extracts phosphorylate the *Sinorhizobium fredii* HH103 effector NopL**. (**a**) In vitro phosphorylation of HH103 NopL by soybean root extracts. A total of 20 μg of root extracts was mixed in a phosphorylation reaction with 5–10 μg of the fusion protein carrying a GST-tag. The empty vector was used as a control. Samples were separated by 12% SDS-PAGE and proteins were visualized by autoradiography. (**b**) Effect of different kinase activity inhibitors on the capacity of soybean root extracts to phosphorylate HH103 NopL in vitro. Samples were separated by 12% SDS-PAGE and proteins were visualized by autoradiography. The molecular weights (kDa) of the protein marker are shown on the left. The control indicates the maximum phosphorylation rate (100%); values under 50% marked with asterisks (*) were considered as phosphorylation inhibition.

**Figure 2 plants-12-02133-f002:**
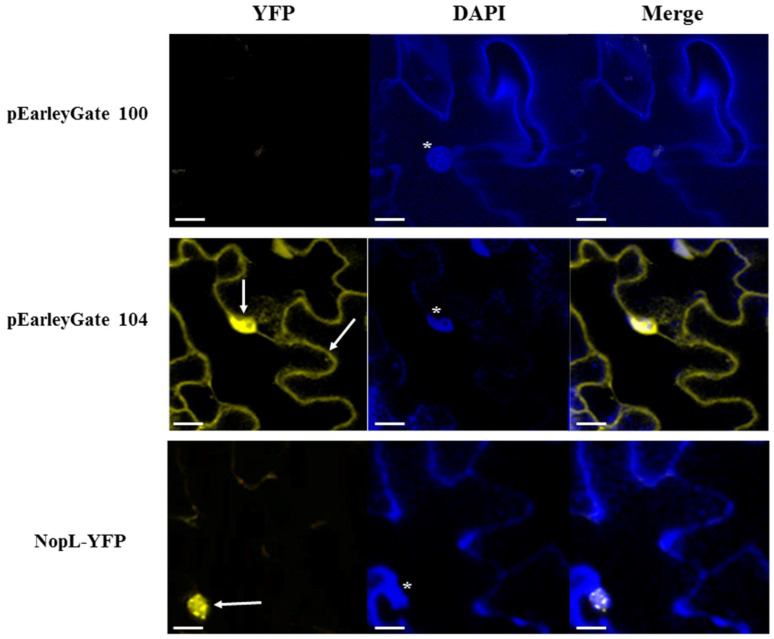
**Figure 2. NopL subcellular location in *N. benthamiana* leave cells.** Arrows indicate NopL-YFP concentrated in nuclear bodies. In the positive control vector pEarleyGate 104, the yellow fluorescence is spread in the cytosol and the whole nucleus, and in the negative control vector pEarleyGate 100, the yellow fluorescence is absent. The plant cell nuclei are stained in blue with DAPI and are marked with asterisks. White bars are scaled to 20 µm.

**Figure 3 plants-12-02133-f003:**
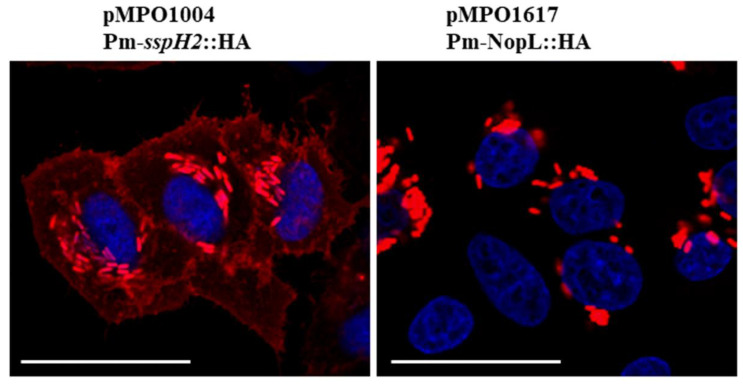
**Salicylate induced HA secretion in Salmonella through the T3SS.** HeLa cell cultures were infected with Salmonella wt strain bearing NopL fused to HA epitope (pMPO1617) or with a secretion control plasmid containing HA epitope fused to sspH2 amino-terminal sequence (pMPO1004). Secretion of the HA epitope is only detected in the control (left panel) but not in NopL-HA (right panel). HeLa cell nuclei were stained with Hoechst (blue) and anti-HA epitope antibody was immunodetected (red). White bars are scaled to 25 µm.

**Figure 4 plants-12-02133-f004:**
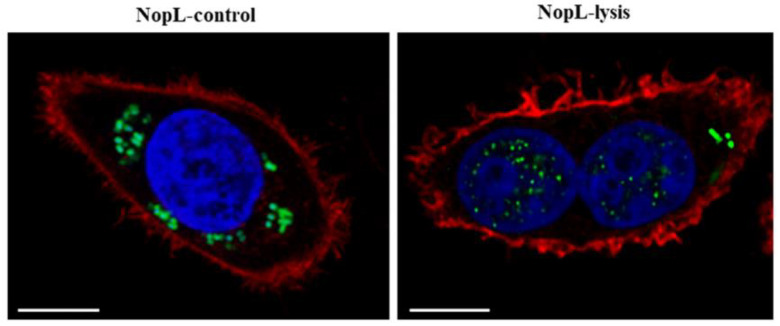
**NopL production and release into HeLa cells cytosol.** HeLa cell cultures were infected with Salmonella sifA^-^ mutants bearing NopL fused to HA epitope (pMPO1617) expression and control (pMPO1631) or lysis plasmid (pMPO1632). NopL was overproduced upon salicylate induction for 4 h and the content of the bacteria was released (right panel) or not (left panel) in HeLa cytosol following an AHT induction. HeLa cell nuclei were stained with Hoechst (blue) and actin filaments (red) with rhodamine phalloidin. Anti-HA epitope antibody was immunodetected (green). White bars are scaled to 10 µm.

**Figure 5 plants-12-02133-f005:**
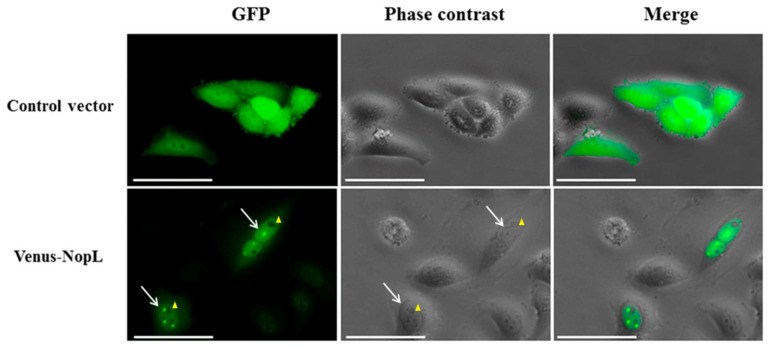
**Venus-NopL transient expression in HeLa cells.** HeLa cells were transfected with pcDNA5/FRT/TO-Venus-Flag (865) control plasmid (upper panels) or with pMPO1643 containing a Venus-NopL fusion (bottom panels). The Venus reporter gene with or without NopL was expressed from pCMV-TetO_2_ promoter upon AHT induction. After 24 h, the green fluorescence was concentrated in nuclear dots in Venus-NopL while in the control vector, it was spread throughout the whole cell. White arrows indicate nuclei borders, and yellow arrowheads show the location of nucleolus. Columns correspond to GFP, phase contrast, and the merge. White bars are scaled to 25 µm.

## Data Availability

Date is contained within the article and Appendix A.

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
