# Peer review of "Multitask Approach to Localize Rhizobial Type Three Secretion System Effector Proteins Inside Eukaryotic Cells"

_plants, 2023, doi:10.3390/plants12112133_

Round 1
Reviewer 1 Report
This work uses microscopic approaches to localize the type III secretion system (T3SS)-dependent effector NopL of Sinorhizobium fredii HH103 in Nicotiana benthamiana leaves and HeLa cell cultures. These methodologies are proposed to address the translocation of T3SS-dependent effectors in both animal and plant eukaryotic cells.
The work is well presented and the results are consistent. I have some minor comments and suggestions.
Comments and suggestions
Line 98, mention that SIPK stands for Salycilic acid-induced protein kinase.
Line 111, include a reference after "plant cell nucleus".
Line 123, It can be indicated if the 4 phosphorylable serine residues are conserved in S. fredii NopL.
Line134, can you discuss and add, if any, some reference as to why Ca2+ is necessary? Is there any reference to calmodulins participating in the phophorylation pathways, as suggested in line 153?
Figure 1: Although it can be deduced from the text, the legend of Figure 1b does not specify what the asterisks and the values 66, 37., 38.... correspond to.
Can the pEarleyGate 100 control be included in Figure 2?
Figure 5 should explain what the images in the three columns correspond to.
Line 312, Is there any domain that indicates that NopL goes to the nucleus?
Line 329 Could you elaborate a little more on the difficulties of localizing NopL-YFP in soybean nodules induced by S. fredii?
A reference/s can be included after " ....epi-fluoresce studies", line 332
A reference/s can be included after " ....compartments", line 337
Minor points:
Lines 196, 218 Salmonella (italics)
Author Response
This work uses microscopic approaches to localize the type III secretion system (T3SS)-dependent effector NopL of Sinorhizobium fredii HH103 in Nicotiana benthamiana leaves and HeLa cell cultures. These methodologies are proposed to address the translocation of T3SS-dependent effectors in both animal and plant eukaryotic cells.
The work is well presented and the results are consistent. I have some minor comments and suggestions.
Thanks for your comments. We will try to address them point by point.
Comments and suggestions
Line 98, mention that SIPK stands for Salycilic acid-induced protein kinase.
Included now, line 100.
Line 111, include a reference after "plant cell nucleus".
Reference 22 has been included, line 114 now.
Line 123, It can be indicated if the 4 phosphorylable serine residues are conserved in S. fredii NopL.
We have changed the sentence to this one: “This effector protein is substrate of plants kinases and phosphorylable residues are conserved between NGR234 and HH103 NopL proteins” Line 127
Line134, can you discuss and add, if any, some reference as to why Ca2+ is necessary?
Regarding to this sentence “In all cases, the presence of Ca2+ in the phosphorylation reaction was essential for the successful phosphorylation of NopL”, we consider it is totally necessary to position Ca2+ as an indispensable molecule for those phosphorylable reactions, since we did not obtain any phosphorylation signal in previous assays performed without it. This is not surprising, since it is well described that Ca2+ signals play an important role in a wide range of the plant processes, such as the response of biotic and abiotic stress, or even it is essential for rhizobia-legumes symbiosis. We include now some references in the text [24-26]. Line 141.
Is there any reference to calmodulins participating in the phophorylation pathways, as suggested in line 153?
Yes, we include in the revised manuscript two references [24,27].
Figure 1: Although it can be deduced from the text, the legend of Figure 1b does not specify what the asterisks and the values 66, 37., 38.... correspond to.
This explanation has been now included in the legend of Figure 1b: The control indicates the maximum phosphorylation rate (100%), values under 50% marked with asterisks (*) were considered as phosphorylation inhibition.
Can the pEarleyGate 100 control be included in Figure 2?
This control is now included, and the figure legend has been amended. The figure has been also modified in base to the comments of another reviewer.
Figure 5 should explain what the images in the three columns correspond to.
The figure has been modified as suggested and also the text referring to this figure and the figure legend.
Line 312, Is there any domain that indicates that NopL goes to the nucleus?
So far there is not any known specific domain for the NopL nucleus translocation (Ge et al 2016).
Line 329 Could you elaborate a little more on the difficulties of localizing NopL-YFP in soybean nodules induced by S. fredii?
Thanks for the suggestion. What we really want to highlight is the difficulty of localizing an effector (NopL-YFP in this case) not exactly in a nodule, but in the root during the infection process. This kind of experiments are tracked in vivo and the cortex of the roots (in soybean) show autofluorescence that hampers the visualization of fluorescent fusion proteins. In addition, the system is often destructive and further analysis of the sample is missed. Furthermore, the low probability to find an event of protein translocation together with the low amount of effector protein secreted, obligates to the use of overexpression systems. Therefore, alternative plant models can be used as detailed in the next lines. Now we include the following clarification. “In the specific case of rhizobacterial effectors, the tracking of its translocation to the root cells of their natural hosts is not always feasible due to the root autofluorescence and the inherent difficulty of handling each cognate partner of such soil bacteria”.
A reference/s can be included after " ....epi-fluoresce studies", line 332
Three references have been included [46, 47, 48]
A reference/s can be included after " ....compartments", line 337
One reference have been included [50]
Minor points:
Lines 196, 218 Salmonella (italics)
Corrected
Reviewer 2 Report
strong methods, good investigation. The idea behind is very interestin
Author Response
Strong methods, good investigation. The idea behind is very interestin
Thank you very much.
Reviewer 3 Report
The manuscript describes the study of a T3SS effector NopL of a rhizobial species Sinorhizobium fredii for its related phosphorylation by plant extract and subcellular localization. The cross kingdom expression of the protein could be innovative, however the manuscript is let down by the quality of data and the ambiguous aim of the study. For example, the purpose of using Salmonella and Hela system to express the proteins instead of using plant-related system such as tobacco BY2 cells or Arabidopsis protoplasts.
Main concerns:
Figure 1a- requires protein control without any extract to suggest the signals are related to phosphorylation but not noise.
Figure 2- the magnification of the microscopic images is low and cannot clearly show the area of nuclei. In addition, DAPI staining is also unclear to show area and overlapping with the signal. There is also no information of the number of cells with the similar observation (statistical analysis) .
Figure 3- the difference of the two images is difficult to comprehend.
Figure 4- it is difficult to be persuaded that the T3SS of Salmonella and Sinorhizobrium are compatible especially the systems are also different. The rationale of using this model is questionable. Moreover, lack of control such as T3SS mutant, making hard to obtain useful conclusion from the images.
Figure 5 - was purposed to support other data, however, the design was incomplete, lacking the necessary controls such as nuclei stain that used in Figure 4. Again, using Hela cells to show the localization of a plant bacteria is not making too much sense, for the cell architecture between plant cells and animals cells is so different including the cell wall/apoplast, organelles such as chloroplast.
Author Response
The manuscript describes the study of a T3SS effector NopL of a rhizobial species Sinorhizobium fredii for its related phosphorylation by plant extract and subcellular localization. The cross kingdom expression of the protein could be innovative, however the manuscript is let down by the quality of data and the ambiguous aim of the study. For example, the purpose of using Salmonella and Hela system to express the proteins instead of using plant-related system such as tobacco BY2 cells or Arabidopsis protoplasts.
Thanks for your comments. We will try to address them point by point.
Main concerns:
Figure 1a- requires protein control without any extract to suggest the signals are related to phosphorylation but not noise.
Thank you for the suggestion. This control has been now included in figure 1.
Figure 2- the magnification of the microscopic images is low and cannot clearly show the area of nuclei. In addition, DAPI staining is also unclear to show area and overlapping with the signal. There is also no information of the number of cells with the similar observation (statistical analysis).
Thank you for the suggestion. The magnification has been slightly increased, and we consider that this new one is optimum to still observe the cytoplasm of the cell where the accumulation of the yellow fluorescence is detected in the positive control, the dimension of this kind of cells is massive. The image has been processed to highlight the nucleus of the cell, and to indicate their position we have includes asterisks. However, DAPI staining in our hands do not always works properly with these samples, and the colouring of the nuclei sometimes is weak and to get a good image we have to overexpose the sample to the confocal laser beam, for this reason the autofluorescence of the cell hinders the appreciation of nuclei borders. This is one of the reasons why we have used a heterologous system as HeLa cells, just to check if we have similar observations. We have found numerous tobacco cells with accumulation of the YFP in the nucleus, it is a frequent event, but we did not perform statistical analysis since our purpose was only detect its presence in enough cells as previously reported by Ge et al. However, the experiment was repetitive and now we include the following information: “Several leaves from independent plants were infiltrated with each condition and numerous cells with yellow dots in nuclei were observed”.
Figure 3- the difference of the two images is difficult to comprehend.
See below.
Figure 4- it is difficult to be persuaded that the T3SS of Salmonella and Sinorhizobrium are compatible especially the systems are also different. The rationale of using this model is questionable. Moreover, lack of control such as T3SS mutant, making hard to obtain useful conclusion from the images.
Dear reviewer, we wonder if these two observations are crossed since in Figure 4 the experiments are independent of T3SS. This is the case of the Figure 3. Therefore, we will try to address these two points in the same answer to solve your concerns. The rationale of trying to induce via Salmonella the delivery of a rhizobial effector into a HeLa cell (what indeed is the real objective of this proposal, independently of the way to), was derived from the report “Secretion of predicted Inc proteins of Chlamydia pneumoniae by a heterologous type III machinery” from Subtil et al. Mol. Micro. 2001. Here, the authors described the heterologous secretion of proteins from Chlamydia using the Type 3 secretion machinery of Shigella flexneri. These bacteria belong to two distinct phyla and harbors distant T3SSs (SctN and SpaL types respectively) as reported by Troisfontaines and Cornelis in 2005. For that reason, we tried to induce the secretion of rhizobial NopL by Salmonella T3SS, two proteobacteria harboring HrcN and SsaN based T3SS, two systems that of course are not so similar as you point out, but they are not as far as those tested by Subtil et al. Additionally, we had the tools to carry out this experiment in a simple way to discard the putative recognition of a conserved signal peptide in NopL by Salmonella T3SS, but unfortunately it didn’t work and so we considered it as a negative result, what support the need to the next steps. Therefore, we do not think that using the control of a T3SS mutant were strictly needed since if the wild type cannot secrete the protein, of course the T3SS mutant neither would. This is what we do show in the Figure 3, where by using the control strain that we published in Medina et al. 2011, we induced and detected the secretion of a HA epitope fused to a signal peptide of a Salmonella effector in the HeLa cell cytosol (left panel), while we were not able to detect HA fussed to NopL in the HeLa cell cytosol with the same method (right panel), probably due as you question, to the differences between these two systems. I sincerely hope that this explanation makes the interpretation of Figure 3 clearer.
However, as we got a negative result, this prompted us to use an alternative approach independent of the T3SS, to release the content of the cell (this is, among additional content from Salmonella cytosol, a huge amount of NopL overproduced) to the cytoplasm of HeLa cells, which definitively is the idea that we wanted to check. Therefore, we used a lysis system to deliver an overproduced heterologous protein into the cytoplasm of a eukaryotic cell that we previously have used (Camacho et al., 2016). With this approach, the result obtained was similar to that found previously in N. benthamiana leaves, and it was independent of the secretion system.
Figure 5 - was purposed to support other data, however, the design was incomplete, lacking the necessary controls such as nuclei stain that used in Figure 4. Again, using Hela cells to show the localization of a plant bacteria is not making too much sense, for the cell architecture between plant cells and animals cells is so different including the cell wall/apoplast, organelles such as chloroplast.
Thank you for this comment, maybe we should have included in the text further information regarding the intracellular lifestyle of some proteobacteria as Sinorhizobium, Salmonella, Brucella or Bartonella (strictly related to rhizobia) among others. It has been previously reported that Sinorhizobium meliloti was responsible for the cytoskeleton reorganization not only in Medicago truncatula (its natural host plant) but also in HeLa cells (Marchetti et al., 2012). This reorganization was associated with the inhibition of several Rho GTPases present in different eukaryotes as plants, animals or yeast, supporting the idea that bacteria that spent part of their life inside eukaryotic cells, have evolved similar technics to influence the intracellular environment on its own benefit. More examples can be found in the literature, as this one “Similar Requirements of a Plant Symbiont and a Mammalian Pathogen for Prolonged Intracellular Survival” from LeVier et al., published in Science in 2000, where they described the parallels of a highly conserved putative cytoplasmic membrane transport present in a rhizobial strain, and in Brucella abortus (a mammalian pathogen), that is critical for the maintenance of both bacteria inside its hosts for prolonged periods. The evolution of the strategies of intracellular proteobacteria (pathogens or symbionts in animal or plant cells) have been carefully reviewed several times as in Batut et al., 2004, revealing similar solutions that bacteria have evolved to survive in such hostile environments. Of course, we agree with you that the cell architecture between plant and animal cells are so different, and thus the use of heterologous systems to test the role of a putative effector, have to be taken into consideration depending on the target of such effector. Obviously if the predicted objective of an effector is a chloroplast or the cell wall, animal cells must not be used. However, for central conserved signaling pathways it could be a good choice due to the suitability of experimental assay. For example, the rhizobial effector NopM is a E3 ubiquitin ligase that belongs to the IpaH effector family found in Shigella, Salmonella and Yersinia all of them sharing the same function, the transference of ubiquitin to proteins to be tagged for degradation in the proteasome. This ubiquitin-mediated proteasome-dependent protein degradation is conserved in eukaryotic cells, and yeasts have been used as eukaryotic model to analyze the function of bacterial effectors that modulates its function regardless the effector is from a plant or an animal interacting bacteria [Xin et al., 2012]. We have included this information in the text now.
Thus, according to yours and other reviewers observations, we have included the following information: “The striking parallels between in-host life-styles of rhizobial strains and other mammalian pathogens, in addition to the fact that the human HeLa cells are a widely-used experimental infection model even for plant interacting bacteria [LeVier et al., 2000; Marchetti et al., 2012], led us to use this in-vitro model for which we had solid experience and many molecular tools available.”
Regarding lack of the DAPI in this figure, we did not carried out such staining since we considered so obvious in base to our experience the accumulation of the proteins inside the nuclei, since nuclei borders were clearly visible. Nevertheless, even though that this image should be more illustrative with the aid of an additional blue channel showing only the nuclei, in our experience we consider that the contour line of the nuclei is visible enough in the phase contrast field and corresponds to the green delimitation on NopL-Venus protein inside the cell in the green channel, where even the nucleolus can be appreciated. However, we have processed the image to highlight the nuclear envelope and we have included the following explanation in the figure legend “White arrows indicate nuclei borders, and yellow arrowheads the location of nucleolus. Columns correspond to GFP, Phase contrast and the Merge respectively”. We have numerous pictures that we can support to show that this is a frequent event.
Reviewer 4 Report
Dear authors,
This is an interesting work about the analyse of the type-III secretion systems in legume-rhizobia symbiosis. This work helps to understand how bacteria are able to colonize plant tissue without exhibiting pathogenic syntoms.
The manuscript is well organized, presented and results are consistent with the conclusions.
However some point should be corrected:
-line 31: It is better to use the term "establish symbiosis" than invade. Symbiosis is a two way relationship with recognition between both parts, invasion is a one way interaction when the first part execute it activity and the second part only suffer the consecuence
-line 53: standarize how type III secretion systems is named in the manuscript
-Improve the quality of the figures 2, 3, 4, and 5
-line 196, 218: Salmonella in italics
-I recommend to include which is the reason to use human cells (HeLa cells) in the analysis of NopL protein location and not the use of plant cells or protoplast
Best regards
Author Response
Dear authors,
This is an interesting work about the analyse of the type-III secretion systems in legume-rhizobia symbiosis. This work helps to understand how bacteria are able to colonize plant tissue without exhibiting pathogenic syntoms.
The manuscript is well organized, presented and results are consistent with the conclusions.
However some point should be corrected:
Thanks for your comments. We will try to address them point by point.
-line 31: It is better to use the term "establish symbiosis" than invade. Symbiosis is a two way relationship with recognition between both parts, invasion is a one way interaction when the first part execute it activity and the second part only suffer the consecuence
Thank you for your suggestion, it has been changed.
-line 53: standarize how type III secretion systems is named in the manuscript
It is now indicated all over the text as Type III secretion system except in the tittle that it is presented with as “three” for formality.
-Improve the quality of the figures 2, 3, 4, and 5
Thanks for the observation. Figures quality have been improved to 300 ppi.
-line 196, 218: Salmonella in italics
Corrected
-I recommend to include which is the reason to use human cells (HeLa cells) in the analysis of NopL protein location and not the use of plant cells or protoplast
Thanks, you are right. Nevertheless, we think that this report support with new data the consideration of increase the range of alternative eukaryotes heterologous models to study the function of effectors, and we have included now the following information: “The striking parallels between in-host life-styles of rhizobial strains and other mammalian pathogens, in addition to the fact that the human HeLa cells are a widely-used experimental infection model even for plant interacting bacteria [LeVier et al., 2000; Marchetti et al., 2012], led us to use this in-vitro model for which we had solid experience and many molecular tools available. In fact, even though that HeLa cells are not the natural hosts of rhizobial strains, it has been previously used as heterologous host to analyze the cytoskeletal reorganization that suffer when is infected with such bacteria [Marchetti et al., 2012]. We also include these two references that support our reasoning.
LeVier et al., Science 2000 Marchetti et al., PLoS ONE 2012
Best regards
Round 2
Reviewer 3 Report
The authors have made some improvements and have provided some reasonable explanations for the manuscript. Although there is still room for improvement, the manuscript can still be acceptable in term of providing additional knowledge to the field.
Author Response
Thank you very much!